# Feasibility and acceptability pilot of video-based direct observed treatment (vDOT) for supporting antitubercular treatment in South India: a cohort study

Rashmi Rodrigues,[1,2,3] Suman Sarah Varghese ![ORCID] ,[4] Mohammed Mahrous,[5] Anil Ananthaneni Kumar ![ORCID] ,[4] Mohammed Naseer Ahmed,[4,6] George D'Souza[7]

For numbered affiliations see end of article.

**Correspondence to**
Dr Rashmi Rodrigues;
rashmijr@gmail.com

## ABSTRACT

**Objectives** The objective of this study was to assess the feasibility and acceptability of video-based anti-tuberculosis (TB) treatment adherence support in patients with TB (PwTB) in South India.

**Design** An exploratory cohort.

**Setting** Participants were recruited at the TB treatment centre (direct observed treatment short centre) of a tertiary-level teaching facility in Bangalore, Karnataka, South India.

**Participants** The study enrolled 25 PwTB, with replacement. Adult PwTB who were on drug-sensitive treatment regimens were included, while those who had drug resistant TB were excluded from the study.

**Intervention** Participants received scheduled adherence reminders and were trained to videorecord themselves swallowing their medication via a mobile application. The application was automated to submit these videos for evaluation. Participants were followed up monthly till treatment completion or withdrawal.

**Outcome measures** Adherence rate and acceptability of video-based directly observed treatment (vDOT).

**Results** The mean±SD age of the participants was 33±14 years, majority were females (16, 64%), residing in urban areas (24,96%), married (17, 68%) and had access to smart phones (23,92%). A total of 3193 person days of follow-up was completed; of the videos submitted within the first 6 months of enrollment (2501), 94% (2354/2501) were considered 'acceptable' and 16 (64%) participants were optimally adherent (ie, ≥80%). Participant videos improved in quality and a higher proportion met acceptability criteria over time. Twenty-one (84%) participants stated that they found the application easy to learn; 13 (52%) preferred vDOT over DOT. Mixed model logistic regression showed that those who are married are more likely have daily adherence to anti-TB treatment.

**Conclusion** Video-based mobile phone interventions are acceptable to PwTB and the ease of using the application increases with time. To provide patient-centred care, vDOT is a promising option that can be offered to patients for treatment support and adherence monitoring.

## INTRODUCTION

Globally, 6.4 million people were diagnosed with tuberculosis (TB) in 2021 with

## STRENGTHS AND LIMITATIONS OF THIS STUDY

⇒ Most study participants were women indicating that women are accepting of digital adherence technologies if given an opportunity.

⇒ Videos not received were equated to missed doses, thereby underestimating treatment adherence, as participants may missed video recording an ingested dose.

⇒ As this was a cohort study with a single group, Hawthorne effect may have played a role in increasing adherence under observation. However, the purpose of video-based directly observed treatment (vDOT) is 'observation', given the varied implementation of directly observed treatment.

⇒ As this was an exploratory pilot, no sample size was calculated, limiting its power, the results of the study should be seen in this light. Yet the results provide valuable information regarding the feasibility and acceptability of vDOT.

India, contributing to approximately one-third of these cases.[1] This, despite the availability of anti-tuberculosis treatment (ATT), comprising antimicrobials and evidence-based preventive strategies.[2]

India has an annual TB incidence of 164 per 100 000 population, of which 87% are newly diagnosed, 10% previously treated and 2.8% multidrug resistant patients with TB (PwTB). As per Indian TB guidelines, ATT generally lasts 6 months and approximately 80% of PwTB are successfully treated annually.[3] Further, 10% of PwTB are categorised as lost to follow-up,[3] non-adherence to ATT, including lost to follow-up is the leading cause for the emergence of drug resistance—impacting costs of care, treatment duration and outcome.[4] Further, TB is distinctive as it impacts the working population and has sizeable economic ramifications.[2]

In order to address the challenges with adherence to ATT, direct observed treatment

short-course (DOTS) was adopted globally as a feasible solution in 1994.[5] Direct observed treatment (DOT), the fulcrum of the DOTS strategy, provides the much-needed follow-up as well as definitive evidence that medications were taken in the prescribed dose and manner. DOT, however, is not devoid of challenges. Facility-based DOT requires patients to visit the treatment provider and often results in lost time and wages. On the other hand, community-based DOT runs the risk of compromising confidentiality due to health workers frequently visiting patients, leading to stigma. Further, such a strategy is human resource intensive from a programme perspective. These concerns, have spurred innovative alternatives to DOT,[5] especially as India's mobile phone penetration is 93% (91% rural and 97%).[6]

The digital adherence technology (DAT) currently available to support ATT range from simple short message service (SMS) reminders and video-based directly observed treatment (vDOT) to sophisticated technologies with ingestible sensors embedded within pills.[7] Further, the Indian National TB Elimination Programme (NTEP), while adopting a patient centred approach in 2014,[8] also included mHealth solutions for adherence support. Some of these mHealth solutions are (1) the 'Nikshay Sampark' call–centre which uses interactive voice response and SMS as adherence reminders,[7] (2) the 'real time medication event reminder monitor,'—an electronic pill box with pressure sensors, that detects the weight of the pills in the box adherence prompts to patients based on the weight of the pill box[9–12] and (3) the 'Patient Compliance toolkit', an mHealth application with a smart card, which delivers adherence support comprising medication reminders, TB health tips, follow-up calls and financial incentives.[7]

'99DOTS' is another mHealth adherence support strategy that the NTEP, rolled out for patients with HIV-TB coinfection in 2015. The strategy subsequently covered all PwTB in 2021 during the COVID-19 pandemic, when DOTS posed a logistic challenge.[13] 99DOTS requires patients to give a missed call to a toll-free phone number that is revealed when the pills are popped from the ATT blister pack.[14] The strategy minimises travel, ensures privacy and supports remote monitoring.[15] The value of these digital innovations for adherence support was realised during the COVID-19 pandemic, when DOT was impossible and remote monitoring the only viable option.

However, while most of these strategies support treatment adherence, video-based DOT (vDOT) is the only strategy that provides definitive evidence that the patient has in fact consumed the medication in the prescribed manner. Given the ubiquity of mobile phones, adherence support for ATT via mobile phones, especially vDOT is a suitable alternative to DOT.[12] In this context, our study provides valuable evidence on the feasibility and acceptability of a video-based treatment adherence monitoring system (vDOT) among patients with pulmonary TB on ATT in South India.

## METHODS

This cohort study piloted the video DOT adherence support intervention for ATT at a private, tertiary care, teaching hospital in Bangalore, South India. The participants for this study were enrolled consecutively from those PwTB who were referred to the DOTS centre for treatment between July 2017 and November 2018. As this was an exploratory study, no sample size was calculated. The participants were recruited comprised newly diagnosed patients with drug sensitive TB, aged ≥18 years, with at least 2 months of treatment pending at enrolment. Participants recruited had to be willing to upload videos of themselves swallowing their ATT via the vDOT mobile application daily. PwTB who were ≥60 years, with HIV coinfection, recurrent TB or drug resistant TB were excluded from the study. Recruitment was done with replacement. Owning a mobile phone was not a necessity as participants were provided a mobile phone for the purpose of the study.

### Study setting

St. John's Medical College Hospital (SJMCH), the study site, is a private, not-for profit, tertiary level, teaching, healthcare facility. The hospital provides emergency, outpatient, inpatient, community outreach and day care services through its 1350 bedded hospital and outpatient facility. It is equipped with a network of laboratories that provide TB diagnostics such as smear microscopy, mycobacterial culture and nucleic acid amplification test (GeneXpert) as well as radiography. Patients diagnosed with TB at the hospital are referred for treatment to the TB treatment centre (DOTS centre) at the hospital itself. The DOTS centre functions through a public–private partnership with the NTEP. PwTB receive treatment from the NTEP at the DOTS centre gratis. Alternatively, PwTB at SJMCH can also avail treatment for a cost at the hospital's pharmacy.

### Standard care

PwTB at SJMCH are treated in accordance with Indian TB treatment guidelines. Patients with drug sensitive TB receive daily treatment based on their weight. The treatment comprises 2 months of an intensive phase with four drugs (isoniazid (H), rifampicin (R), pyrazinamide (Z), ethambutol (E)) followed by 4 months of a continuation phase with three drugs (H, R, E). This treatment is available in fixed dose combinations.

Conventionally, for ATT adherence support, PwTB are expected to visit a healthcare provider 6 days a week and swallow their medications under direct supervision, but may be practised variably. These visits are documented on a treatment card. Each PwTB receives a minimum of 180 days of treatment. If a dose (day) is missed, the treatment continues till the required number of doses are completed. While, adherence monitoring at the time of the study, was via in-person DOTS, currently, all PwTB receiving ATT through the NTEP receive 99DOTS.

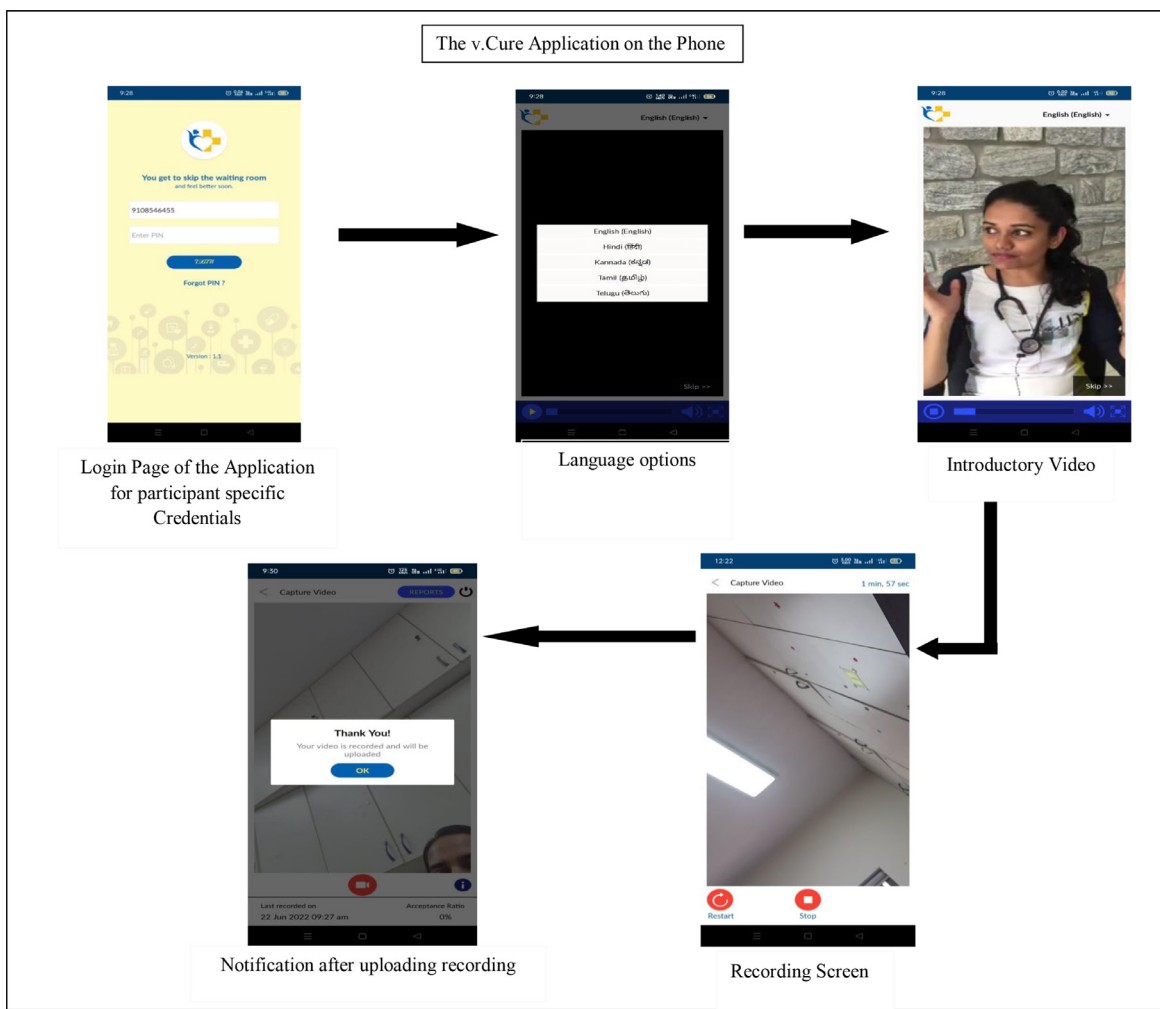

The v.Cure Application on the Phone

Login Page of the Application for participant specific Credentials

Language options

Introductory Video

Notification after uploading recording

Recording Screen

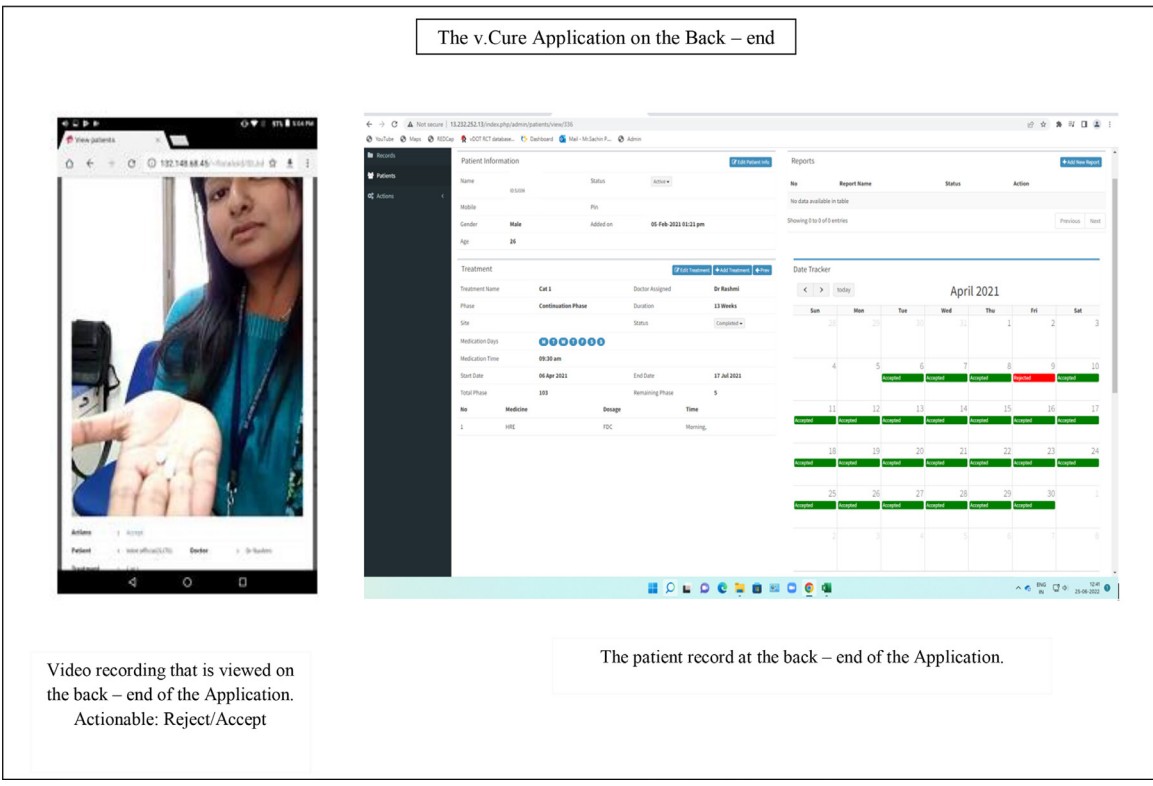

The v.Cure Application on the Back – end

Video recording that is viewed on the back – end of the Application. Actionable: Reject/Accept

The patient record at the back – end of the Application.

**Figure 1**  v.Cure application operation.

## The intervention: vDOT

vDOT, PwTB require to videorecord themselves taking their medications via a mobile application, which then sends the videos to the study team via the internet. All participants also receive an adherence text reminder via the application daily at the time of their choice. For the purpose of this study, the study team developed a vDOT mobile application the v.Cure, installed on the participants' mobile phones.

Trained research assistants (not routine health staff) reviewed the uploaded videos at the provider interface of the application, and classify the videos as 'accepted' (dose taken) or 'rejected' (video received but dose not taken). The videos were accepted if the participant was identified, all the pills were shown before swallowing under adequate lighting. If not, the video was 'rejected'. A participant is considered to have 'missed' videos and thereby their ATT dose if no video was received for the day. Participants who missed uploading videos were contacted via their mobile phone on the subsequent day by the study team.

### The vDOT mobile application: v.Cure

The application comprised two interfaces: (1) the user interface and (2) the provider interface (figure 1).

User interface: The v.Cure is available at the Google Play Store in five languages, that is, English, Kannada, Hindi, Tamil and Telugu, each in their own script (https://play.google.com/store/apps/details?id=in.stjhons). The application is restricted to users in India. Only those registered by the healthcare provider can use the application. Once registered, the participant may log into the application installed on their mobile phones, using their credentials. The participant then views an instructional video within the application that demonstrates how to capture and upload the adherence video. Participants are free to skip the instructional video and access the recording screen directly to record the video. After the recording, the participant is informed of the status of the video, that is, pending/uploaded. This is an automated process. The participant may choose the language for the application as well as the time for their adherence reminder notification at registration.

Provider interface: The provider interface has the demographic and clinical profiles of the participants, as well as their submitted video linked to their adherence calendar. The research staff view the video and indicate if the video is accepted or rejected, while the application itself flags a missed video. The adherence calendar reflects the status of the videos (accepted, rejected, missed), and also communicates the status to the participant via the v.Cure application.

### Participant enrolment and follow-up

At baseline, trained research assistants administered consent, obtained relevant demographic and clinical information and installed the v.Cure mobile application onto the participants mobile phone. If the participant did not own a mobile phone but wished to participate in the study, a mobile phone was provided. The participant was then trained to use the mobile application and upload videos of themselves taking their medication. In case a participant as unable to use the application, a family member was chosen and trained to record and upload videos of the participant.

Study follow-ups were monthly and combined with pill refill visits, pill counts and clinical assessments. The v.Cure adherence calendar helped monitor adherence and reasons for missed videos were obtained via phone calls or at follow-ups. Acceptability of the intervention was assessed at the end of the study. Each participant received INR500/month for internet, time, as well as study related travel.

## Study outcomes

Daily adherence at individual participant level: If a video is accepted the participant is considered adherent, while if missed or rejected participant is considered non-adherent.

The 'adherence rate' is the proportion of accepted videos to the total number of videos expected (includes accepted, rejected and missed). 'Optimal adherence' for the purpose of this study is defined as an adherence rate ≥80%.[16]

## Intervention fidelity

All the participants who fulfilled the selection criteria and consented to participate in the study were trained in the use of vDOT application—registration, logging in, recording and uploading videos. Participant concerns regarding these procedures were addressed at registration and throughout the study duration. Further, phone numbers of study personnel were provided to participants to call in the event of a challenge with the application. The research team ensured that all the videos that were uploaded daily were viewed promptly. Participants who missed videos or submitted erroneous videos were contacted and their concerns recorded and addressed. Issues with application identified on the go were communicated with the technical team and addressed swiftly.

## Analysis

Data comprising participant demography and clinical features obtained at baseline and follow-up were entered in Microsoft Excel. These data were combined with the adherence from the v.Cure application, also in Microsoft Excel. STATA V.12.0 was used for analysis and the data described as frequencies and measures of central tendency and deviation. Participant experiences with the vDOT intervention were similarly described.

A multilevel mixed models logistic regression analysis was done with daily treatment adherence as the dependent variable. The fixed effects model includes participant demography and clinical factors as independent variables. The random effects model has two levels, the first was daily adherence, and at the second the individual participant (figure 2). Adjusted ORs (95% CI), intraclass

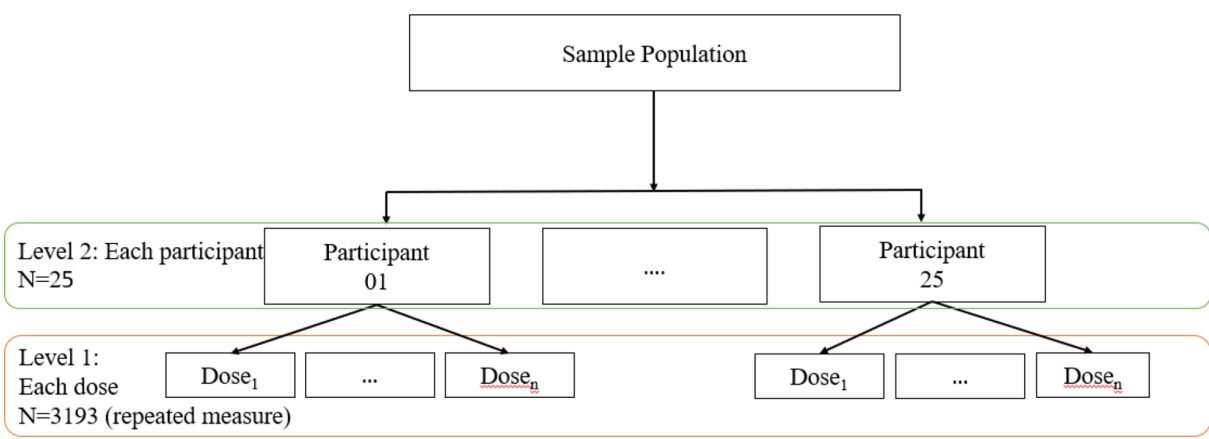

**Figure 2** Schematic presentation of the levels in the hierarchical model.

cluster (ICC) coefficients and Akaike information criterion) are reported.

In addition, a mixed models logistic regression where the binomial dependent variable was daily participant responsiveness. Responsive meant video accepted or rejected or participant responded to follow-up call, non-responsive meant participant did not respond to a follow-up call (figure 2; online supplemental annexure table 1).

### Patient and public involvement

This was a part of a larger trial that uses vDOT as an adherence monitoring and treatment support tool (CTRI no: CTRI/2017/07/009052). Prior to the development of the intervention, acceptability of such an intervention among PwTB was conducted in Bangalore, South India and in Ujjain, Central India.[17 18] Patients' suggestions and concerns were taken into consideration during the design of the trial and during the development of the v.Cure application.

## RESULTS

A total of 258 patients were screened, of whom 150 were eligible, and 40 were initially enrolled in the study. Eligible male participants (62, 56%) who refused consent cited 'time constraints' while eligible female participants (48, 44%) cited privacy and stigma-related concerns as reasons for refusal. The median time to enrolment from treatment initiation was 1 day (IQR 0–15 days) with a maximum of 109 days. The most common reasons for ineligibility were age <18 years or >60 years (40, 37%),

**Table 1** Reasons for replacing participants after enrolment and for refusing consent

| Sl. no. | Reasons cited | No (%) |
|---|---|---|
| **Unwilling to participate** | | **N=110** |
| 1 | Not willing to learn to use a smartphone | 37 (34) |
| 2 | Not interested in research | 20 (18) |
| 2 | Stigma and privacy issues | 22 (20) |
| 3 | Time constraints | 14 (13) |
| 4 | Other health issues | 6 (5) |
| 5 | Family not comfortable | 4 (4) |
| 6 | Do not believe they have TB | 3 (3) |
| 7 | Would need to depend on others | 2 (2) |
| 8 | Network issues at home | 2 (2) |
| **Participants who were replaced** | | **N=15** |
| 1 | Withdrew consent after participants' families were concerned of stigma and privacy | 5 (33) |
| 2 | Participants were unable to upload videos | 3 (20) |
| 3 | Participants were unable to find a family member to either take videos or upload them | 3 (20) |
| 4 | Lost to follow—up before 2 months of initiation of the intervention | 3 (20) |
| 5 | Network issues | 1 (7) |

TB, tuberculosis.

**Table 2** Sociodemographic and clinical features

| Factor | Category | Female | Male | Total |
|---|---|---|---|---|
| N | | 16 | 9 | 25 |
| Age, mean±SD | | 32.8±15 | 33.6±12 | 33±14 |
| Family income,* n (%) | BPL | 6 (38%) | 7 (78%) | 13 (52%) |
| | APL | 10 (62%) | 2 (22%) | 12 (48%) |
| Residence, n (%) | Rural | 0 (0%) | 1 (11%) | 1 (4%) |
| | Urban | 16 (100%) | 8 (89%) | 24 (96%) |
| Marital status, n (%) | Single | 6 (38%) | 2 (22%) | 8 (32%) |
| | Married | 10 (62%) | 7 (78%) | 17 (68%) |
| Children, n (%) | Yes | 9 (56%) | 6 (67%) | 15 (60%) |
| | No | 7 (44%) | 3 (33%) | 10 (40%) |
| Owns a phone, n (%) | Yes | 15 (94%) | 9 (100%) | 24 (96%) |
| | No | 1 (6%) | 0 (0%) | 1 (4%) |
| Access to a smartphone, n (%) | Yes | 14 (87%) | 9 (100%) | 23 (92%) |
| | No | 2 (13%) | 0 (0%) | 2 (8%) |
| TB site, n (%) | Pulmonary | 4 (25%) | 6 (67%) | 10 (40%) |
| | Extra pulmonary | 12 (75%) | 3 (33%) | 15 (60%) |
| Median duration between treatment initiation and enrolment (in days) | | 1.5 (0–12.25) | 1 (0–9) | 1 (0–15) |
| Phase of treatment at the time of enrolment | Intensive phase | 14 (87%) | 8 (89%) | 22 (88%) |
| | Continuation phase | 2 (13%) | 1 (11%) | 3 (12%) |
| Prefer vDOT | Yes | 10 (62%) | 3 (33%) | 13 (52%) |
| | No | 6 (38%) | 6 (67%) | 12 (48%) |
| Remained till treatment completed treatment, n (%) | No | 3 (19%) | 0 (0%) | 3 (12%) |
| | Yes | 13 (81%) | 9 (100%) | 22 (88%) |

Withdrew consent: 3, 2nd month 2 (due to family concerns regarding stigma), 3rd month 1 (due to other health-related issues).
*According to WHO standards, BPL—below poverty line, APL—above poverty line.
vDOT, video-based directly observed treatment.

patients with HIV-TB coinfection (23, 21%) and recurrent TB (2 anti-tubercular therapy) (19, 17%).

Not owning/not interested in learning to use a smartphone (37 (34%)) and not interested in research (20 (18%)) were the most common reasons for declining to participate in the study (table 1).

Of the 40 participants enrolled, 15 participants dropped out within 2 months of enrolment and were replaced; the most common reason for withdrawal was concern regarding privacy and stigma (5, 33%). The remaining 25 participants contributed 3193 person days of follow-up to the study.

Demographic characteristics of the 25 study participants who completed at least 2 months of the intervention are given in the table 2. Sixteen (64%) participants were female and 9 (36%) were male. Most participants had an urban background (96%), were married (68%) and had access to a smartphone (92%). The median income of the participants was INR15 000 (IQR INR7000–INR22 750). Nearly equal numbers of participants preferred vDOT (52%) to DOT (48%).

## Adherence via vDOT

vDOT adherence assessment required a minimum of 2 months and a maximum of 6 months within the intervention. During this period in the study we received 78% (2501/3193) of the expected videos from the participants. Of the videos received, 94% (2354/2501) were 'accepted'. The proportion of uploaded videos that met the acceptability criteria increased over time as participants' familiarity with the application grew (figure 3).

The overall average adherence was 70% (95% CI 68% to 87%), and ranged between 12% and 99%. Sixteen (64%) participants were optimally adherent through vDOT of ≥80% (mean adherence (95% CI) 91% (88% to 95%)). The remainder had less than 80% of videos accepted (12 participants, mean (95% CI) adherence 51% (38% to 68%)).

Participants who were female, those married, those literate, those gainfully employed, those below poverty line socioeconomic status, those who were enrolled earlier on in their treatment and those who preferred DOT were more adherent to medication (table 3). However, the analysis lacks power.

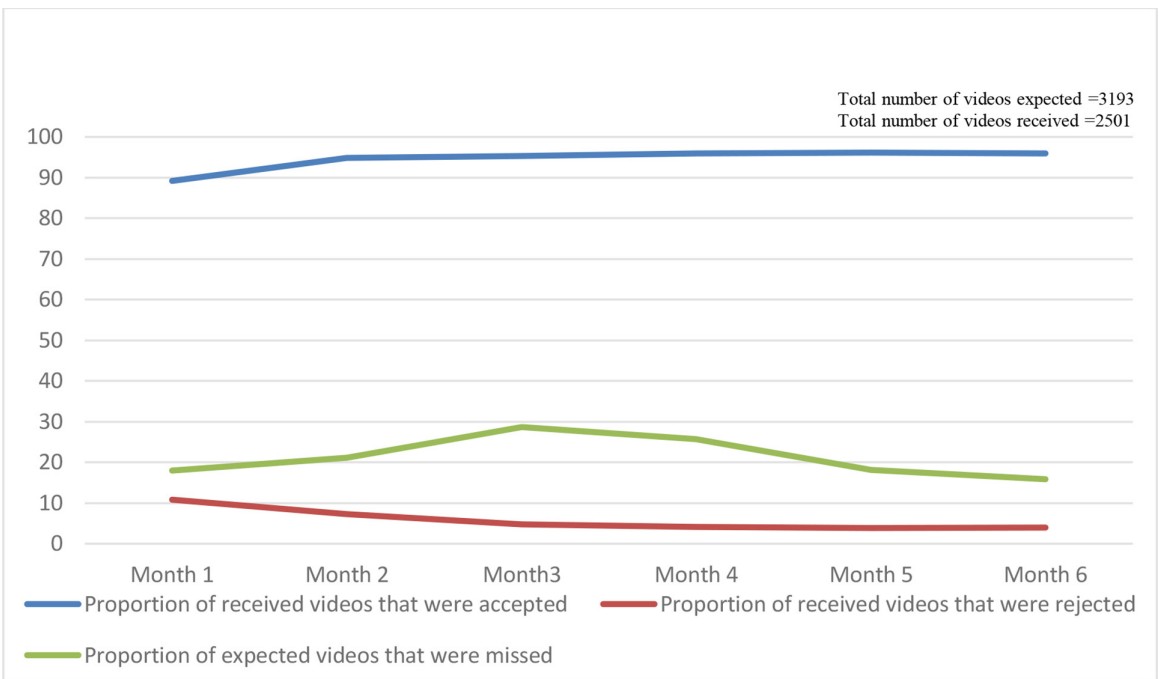

**Figure 3** Proportions of videos that were accepted, missed or rejected over the period of the study.

A mixed model analysis was carried out with daily adherence to ATT as the outcome variable. The ICC of the null model indicated that 33% of the variation was contributed by inter cluster variation. The final model showed that those who were unmarried were more likely to have daily adherent (table 4).

All 25 participants had at least 1 missed video. Of the 692 missed instances, participants could not be contacted in

| Table 3 Comparison between proportion optimally adherent and demography | | | |
|---|---|---|---|
| **Factors** | **Optimally adherent (n)** | **Total N=25** | **Unadjusted OR (95% CI)** |
| Sex | | | |
| Female | 10 (63%) | 16 | 1.20 (0.16 to 12.3) |
| Male | 06/16 (67%) | 09 | Reference |
| Age(mean±SD) years (n=16) | 34±16 | 33±14 | – |
| Employment | | | |
| Unemployed | 07 (64%) | 11 | 0.63 (0.09 to 4.27) |
| Employed | 09 (64%) | 14 | Reference |
| Socioeconomic Status | | | |
| BPL | 09 (64%) | 14 | 1.03 (0.14 to 7.04) |
| APL | 07 (64%) | 11 | Reference |
| Literacy | | | |
| Illiterate | 02 (67%) | 03 | 0.88 (0.01 to 19.52) |
| Literate | 14 (64%) | 22 | Reference |
| Marital status | | | |
| Unmarried | 06 (75%) | 08 | 0.49 (0.04 to 3.94) |
| Married | 10 (59%) | 17 | Reference |
| Time to treatment initiation in days (mean±SD) n=16 | 1.19±0.4 | 1.75±0.33 | – |
| Participant preference vDOT | 08 (62%) | 13 | 1.24 (0.18 to 8.87) |
| DOT | 08 (67%) | 12 | Reference |
| APL, above poverty line; BPL, below poverty line; DOT, direct observed treatment; vDOT, video-based directly observed treatment. | | | |

**Table 4** Mixed-effects multilevel logistic regression analysis of the determinants of daily adherence to ATT

| Variable | | Null model | Logistic regression | Final model |
|---|---|---|---|---|
| **Fixed effects model** | | **Adjusted OR (95% CI)** | **Adjusted OR (95% CI)** | **Adjusted OR (95% CI)** |
| Sex | Females | | 0.44 (0.31 to 0.62) | 0.43 (0.06 to 1.99) |
| | Male | | Reference | Reference |
| Age | | | 0.99 (0.98 to 1.0) | 0.98 (0.94 to 1.03) |
| Area of residence | Urban | | 3.86 (2.02 to 7.38) | |
| | Rural | | Reference | |
| Marital status | Married | | 0.41 (0.30 to 0.57) | 0.42 (0.30 to 0.58) |
| | Unmarried | | Reference | Reference |
| Employment status | Employed | | 0.18 (0.03 to 1.42) | 0.18 (0.02 to 1.43) |
| | Unemployed | | Reference | Reference |
| Family income | Above poverty line | | 1.82 (1.5 to 2.22) | 2.13 (0.70 to 6.52) |
| | Below poverty line | | Reference | Reference |
| Literacy | Literate | | 0.32 (0.20 to 0.52) | 0.35 (0.03 to 3.65) |
| | Illiterate | | Reference | Reference |
| Owns a phone | Yes | | 0.16 (0.99 to 1.0) | 0.13 (0.006 to 2.94) |
| | No | | Reference | Reference |
| Site of TB | Pulmonary TB | | 0.98 (0.78 to 1.23) | 1.07 (0.34 to 3.38) |
| | Extrapulmonary TB | | Reference | Reference |
| Patient Preference of vDOT | Prefers vDOT over DOT | | 0.45 (0.35 to 0.58) | 0.94 (0.23 to 3.77) |
| | Prefers DOT over vDOT | | Reference | Reference |
| Time from treatment initiation to enrolment | | | 1.0 (0.99 to 1.0) | 0.99 (0.94 to 1.03) |
| Random effects model | | | | |
| At individual level | | | | 1.09 (0.81 to 1.47) |
| Intraclass cluster coefficient (95% CI) at Individual level | | 0.34 (0.22 to 0.48) | | 0.27 (0.17 to 0.40) |
| Akaike information criterion | | 2966 | 3457 | 2978 |

ATT, anti-tuberculosis treatment; DOT, direct observed treatment; TB, tuberculosis; vDOT, video-based directly observed treatment.

nearly half (326) of these instances. A mixed model analysis was carried out with responsiveness as the outcome variable. The ICC for the null model indicates that 70% of the variation is accounted for by intercluster variation. The model indicated that those who were unemployed were more likely to have daily responsiveness during the follow-up period (online supplemental annexure table 1).

For the 460 (66%) missed instances where participants responded to calls, 'technical issues' with the application was cited as the most common reason for not uploading the video (table 5).

### Acceptability and feasibility of vDOT
Majority of the participants reported perfect functioning of the application. Eleven (44%) participants faced some challenges with accuracy and speed of the application features and components. Most participants stated that they were either able to use the application immediately (11 (44%)) or it was easy to use (10 (40%)). Majority of the participants found the menu labels/icons/buttons

to be perfectly clear (14 (56%)). Participants commonly expressed that interactions were perfectly consistent and intuitive across all components and screens (figure 4).

All the participants felt that the arrangement and size of the buttons/icons/menus/content on the screen were satisfactory and appropriate. Over half the participants reported that the resolution of the graphics used was of very high quality. The majority stated that the application had very high visual appeal (figure 4). Less than two-fifths of the participants found the application difficult to manoeuvre once opened.

### Privacy and application related preferences
Most of the participants were not worried about discrimination due to TB (84%). While 2 (8%) participants found vDOT intrusive (8%), most participants found the passcode feature effective in protecting their privacy (20 (80%)). A quarter of the participants stated that the mobile application had been accessed by others in the preceding month and nearly all the participants reported that someone had seen them recording the

**Table 5** Reasons for not uploading (missed) videos among those who were contacted

| Sl. no. | Reasons | Instances, n=460 | Participants N=25 |
|---------|---------|------------------|-------------------|
| 1 | Technical issues—application or phone | 89 (19%) | 15 (60%) |
| 2 | Claimed to have uploaded | 87 (19%) | 17 (68%) |
| 3 | Travel—personal or health related | 76 (17%) | 11 (44%) |
| 4 | Lost phone/phone repair | 33 (7%) | 8 (32%) |
| 5 | Shared phone/dependent | 30 (7%) | 4 (16%) |
| 6 | Forgot to upload video | 25 (5%) | 10 (40%) |
| 7 | No network | 23 (5%) | 10 (40%) |
| 8 | Wants to withdraw | 19 (4%) | 3 (12%) |
| 9 | No power | 18 (4%) | 10 (40%) |
| 10 | Forgot to take medicines | 12 (2%) | 3 (12%) |
| 11 | Change in schedule/phase | 11 (2%) | 5 (20%) |
| 12 | Still learning the app | 15 (3%) | 4 (16%) |
| 13 | Accidentally deleted the app | 8 (2%) | 7 (28%) |
| 14 | In person DOTS | 7 (2%) | 8 (32%) |
| 15 | No permission from DOTS provider | 7 (2%) | 3 (12%) |

DOTS, direct observed treatment short.

video. However, none of these instances were followed by a negative experience. Most participants preferred recording the videos themselves (figure 5).

### Intervention fidelity

The most common reason for a video not uploaded or recorded as uploaded but not received by the study team was 'technical issues with the application' this referred to issues at both, the user and provider interface (89, 19%). These problems (table 6) were resolved after discussion with application developers. An interesting issue identified was incompatibility of the application with a few lower-end smart phone models. This was resolved by adjusting settings in the Google play store, while mobile phone models that continued to be incompatible with the application were not used in the study. Participants with such phones or those without phones were given a smart phone for the purpose of participation.

### DISCUSSION

This study aimed to assess the acceptability and feasibility of vDOT for ATT adherence support. We observed that nearly two-thirds (16, 64%) of the participants had optimal adherence (ie, ≥80%) with vDOT, with no socio-demographic factors influencing adherence. Most participants found the application easy to use, with high visual appeal. Those who belonged to above poverty live were more likely to have daily adherence to ATT.

While the most common reasons for not participating in the study were unwillingness to learn how to use a smartphone and disease related stigma, known barriers to mHealth are poor internet connectivity, security and privacy as well as English literacy, as reported in other studies,[19] also played a role. However, as the application was video based it required participants to learn the skill of opening the application and recording the video. This does not require language literacy. Further, as the participants were largely urban, internet connectivity was not an issue.

Of the 2501 doses received on the application, over 90% were of acceptable quality, While 40% of the participants forgot to record a video at least once during follow-up, only 12% had actually missed taking their medication. A cross-sectional study done among HIV-TB coinfected patients using 99DOTS, reported that 26% had forgotten to take the medication and 6% participants forgot to make a call after taking the medication.[20] This could be attributed to the fact that unlike 99DOT, where the patient gives a missed call after taking the medication, in vDOT the act of taking the medication is linked to recording the video thereby minimising missed instances.

However, for vDOT, if we did not receive a video we equated it to a missed dose. While doing so underestimated the true adherence it likely ensured robustness. This especially, as the viewing every dose taken via the mobile application, is the alternative to DOT that we propose. While this might imply vDOT as a stand-alone intervention is questionable, evidence indicates that DAT such as vDOT holds promise when combined with other interventions such as family/community DOTS or peer support.

Furthermore, mHealth interventions are expected to improve the quality, accessibility and cost-effectiveness of healthcare interventions available to patients.[19] According to the National Strategic Plan for TB (NSP) 2017–2025, a good patient support plan is vital for treatment success and should be developed at the time of treatment initiation.[7] Educating patients, families and communities regarding TB and its treatment while improving access to patient centred care also improves treatment adherence and outcomes.[17] The vDOT application also has the potential to address these needs if tailored to the individual, thereby personalising the intervention.

Being asynchronous, vDOT has similar benefits as 99DOTS and sensor observed treatment (SOT), namely, convenience, decreased absenteeism from work and reduction in travel costs.[18] In addition, logistic difficulties during the manufacturing and supply of customised sleeves for the ATT blister packs used in 99DOT are avoided.[21] vDOT can also minimise the discomfort or extra effort associated with SOT, which requires the patient to

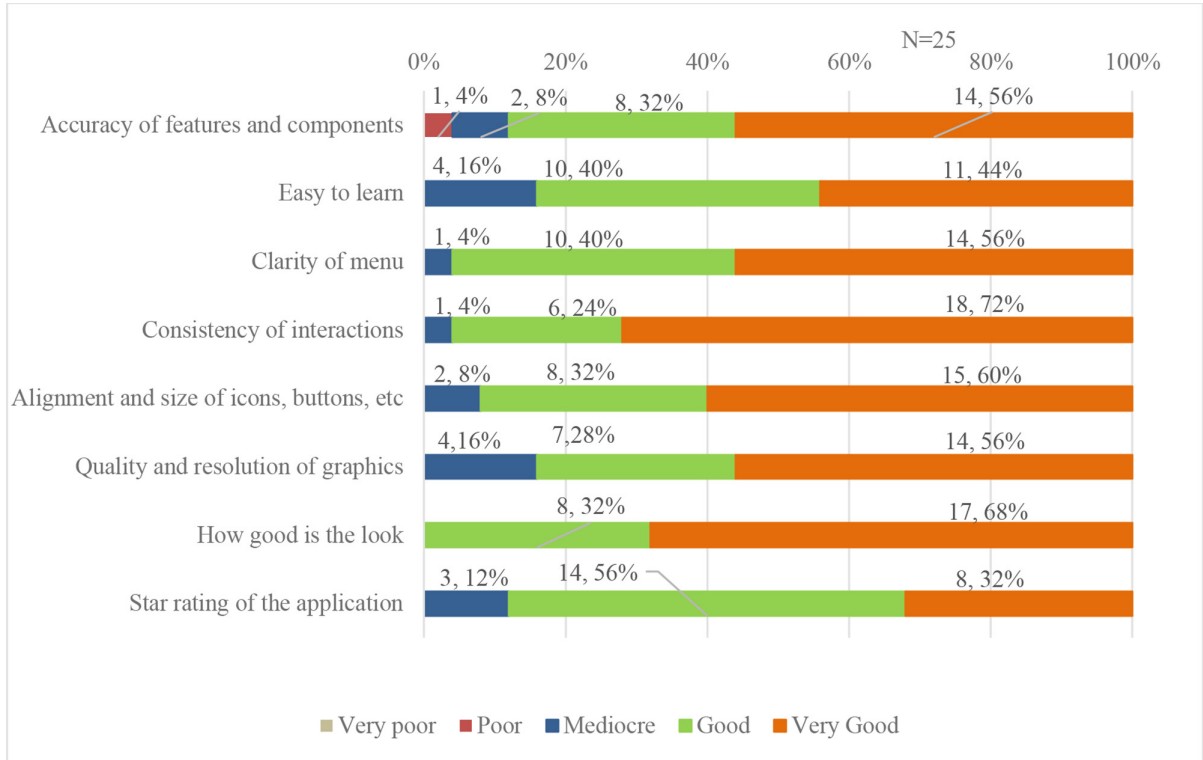

**Figure 4** Acceptability of the application.

wear wireless wearable readers that capture data from the sensor within the ingested pill. The wearable readers are usually worn on their wrist, arm, hip or abdomen, which if seen by others can potentially increase stigma. On the other hand vDOT is considered a less intrusive and a more autonomous adherence monitoring tool[19 20] as the

participant can use an inconspicuous mobile phone for the purpose. In addition, patients may find ingesting a sensor intrusive[22] and the bulk of the wearable sensor is a barrier to its uptake.[23–25]

The vDOT application we developed uses a passcode to authenticate access to the application. When participants

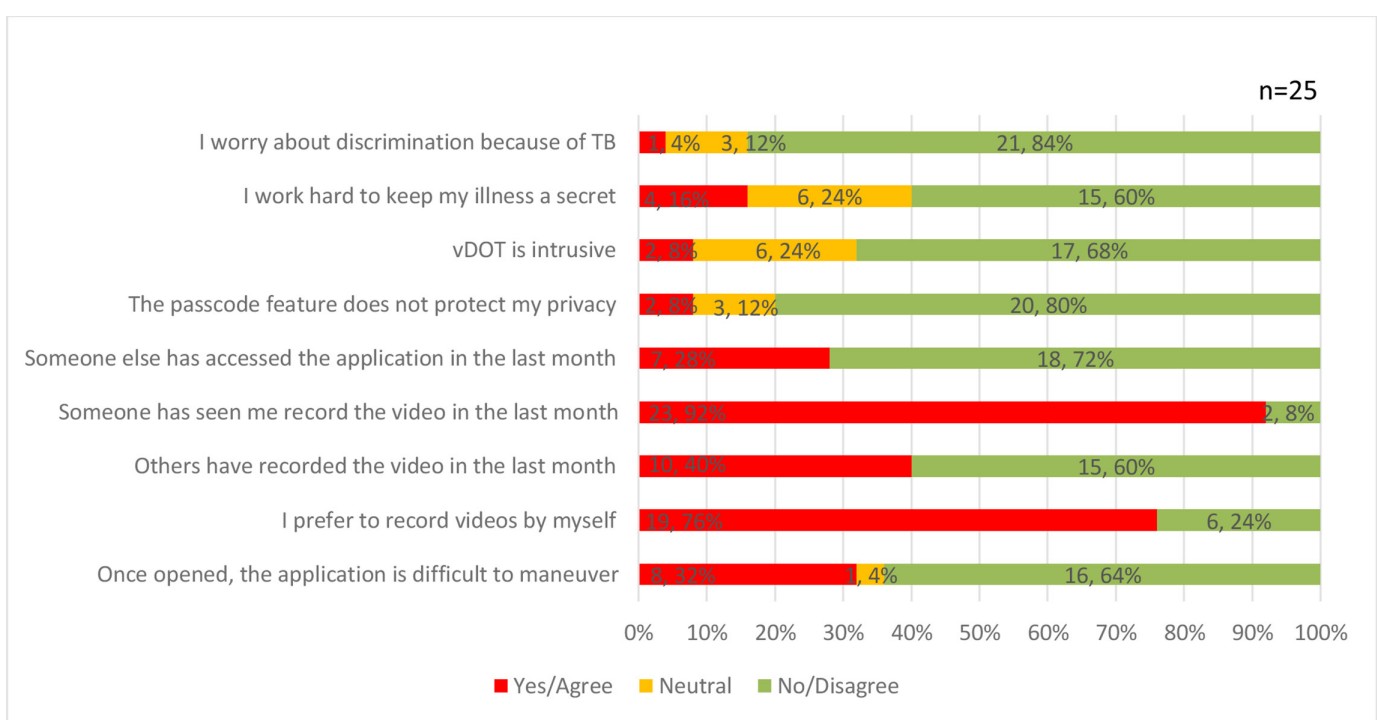

**Figure 5** Concerns regarding privacy and application-related preferences. TB, tuberculosis; vDOT, video-based directly observed treatment.

**Table 6** Issues identified with the vDOT application

| Sl. no. | Issue raised |
|---|---|
| Participant perspective | |
| 1 | The pop-up reminders intended to prompt participants to take medications were not working |
| 2 | In case participants forgot their passcode to access the application, they would contact the team and a new pin would be generated. |
| Research team perspective | |
| 3 | The application was not compatible with two specific smartphone models |
| 4 | The application was designed to automatically calculate the 'acceptance ratio', however, the calculation was inaccurate |
| 5 | Once the video received from the participant was reviewed, the participant list shuffled |
| 6 | The provider interface screen would hang after rejecting a video |
| 7 | At times uploaded videos, could be viewed only after 1–2 days |
| 8 | The missed instances were not sorted in any particular order, we requested the application developers to sort them by date |
| 9 | Missed doses were not captured in the data tracker calendar for individual participants |
| 10 | Percentage adherence, at times, exceeded 100% |

vDOT, video-based directly observed treatment.

forgot their passcode, they called the study team to generate a new code. Aside from concerns regarding the authenticity of the caller, this system worked efficiently to provide participants with timely access to the application. However, we did not receive any complaints from the participants stating that others had accessed their phones and generated passcodes. Further, as the passcodes as well as the user IDs were numeric, only digital literacy was required.

The limitation of the current version of the application is that it does not support the reporting of adverse events (AEs). The decision to exclude this feature was made by the research team to reduce complexity in terms of resources required. Although participants were provided with appropriate contact numbers to reach out to the study team in an emergency. In the future, the application may be modified to include reporting of AEs by patients. The vDOT minimised patient–provider interaction. To overcome this drawback, participants were called whenever a missed dose was recorded, in addition to the monthly visits to the facility.

Interventional fidelity was ensured by uncompromising adherence to the protocol, appropriate training of participants to use the application and timely redressal of challenges identified—which was one of the purposes of the pilot. While patient preferences may depend on familiarity with the mHealth delivery platform,[26] our study did not find familiarity to influence adherence and is likely due to the small sample size. The study also shows that patients with no exposure to DAT can be successfully trained to use them, which was confirmed with the increasing proportion of videos that were accepted increases with time. Overall, while the small sample size limits generalisability, the study has helped identify and address technical challenges with the application and while understanding its acceptability. Also, approximately two-thirds of the participants were women, as many men with TB refused to participate in the study citing employment and data security issues. This implies a selection bias as the proportion of men with TB is higher than that of women and so also their access to digital technologies.[27] However, the valuable insight into the uptake of DAT by women, that the study provides, if given an opportunity, cannot be ignored.

In addition, this was a single arm cohort study that not only excluded participants unwilling to comply with the protocol but also replaced some. While this makes the study more explanatory than pragmatic, it is also subject to the Hawthorne effect, which however, addresses the premise of vDOT that is, improving treatment adherence through observation, as the implementation of DOT may vary.

## CONCLUSION

We found that vDOT is an acceptable alternative to conventional DOT for ATT adherence support and monitoring. The application was easy to learn and non-intrusive. Though concerns with privacy and stigma limited participation, features like the passcode to access the mobile application helped alleviate participant concerns. However, given the increasing digital literacy and the growing confidence in digital technology in the backdrop of pandemics such as the COVID-19, the use of vDOT for adherence support in TB is imperative. Yet, with the availability of multiple digital adherence interventions, PwTB may be offered a basket of choices—similar to the 'cafeteria approach' for contraception, for treatment adherence support—of which vDOT is one.

**Author affiliations**
[1]Community Health, St John's National Academy of Health Sciences, Bangalore, Karnataka, India
[2]Global Public Health, Karolinska Institute, Stockholm, Sweden
[3]Intermidiate Fellow, Clinical and Public Health, DBT/ Wellcome Trust India Alliance, Hyderabad, India
[4]St. John's Research Institute, St John's National Academy of Health Sciences, Bangalore, Karnataka, India
[5]Independent Researcher, Bangalore, Karnataka, India
[6]Blackpool Teaching Hospital, NHS trust, Blackpool, UK
[7]Pulmonary Medicine, St John's National Academy of Health Sciences, Bangalore, Karnataka, India

**Acknowledgements** The authors would like to thank Ms Shubha Krishnamurthy, Ms Hannah DJ, Dr Nikitha Prabhakar and Dr Naseer Ahmed who supported data

collection and data management. The authors also appreciate the support from the offices of the district TB officers, Bruhat Bengaluru Mahanagara Palike and Bangalore Urban District, Bangalore, India, through their respective public health care facilities in urban and rural Bangalore. We would like to thank Focaloid Technologies for their dedication and approachability that was instrumental in the development of the application. We would also like to thank Dr Tinku Thomas for the statistical support provided.

**Contributors** RR and GDS were involved in the concept and design of the study and in monitoring the study, AAK and MNA were involved in administering informed consent, data collection, entry and monitoring, MM and SSV were involve in drafting the paper and data analysis, RR was involved in the final revision of the article. RR is the guarantor for the content.

**Funding** This work was supported by DBT/Wellcome Trust India Alliance (number IA/CPHI/15/1/502042) and the Swedish Research Council (number 2015-02750).

**Competing interests** None declared.

**Patient and public involvement** Patients and/or the public were involved in the design, or conduct, or reporting, or dissemination plans of this research. Refer to the Methods section for further details.

**Patient consent for publication** Consent obtained directly from patient(s).

**Ethics approval** This study involves human participants and was approved by Institutional Ethics Review Board,St. John's Medical College, reference number 342/2016. Participants gave informed consent to participate in the study before taking part.

**Provenance and peer review** Not commissioned; externally peer reviewed.

**Data availability statement** Data are available in a public, open access repository.

**ORCID iDs**
Suman Sarah Varghese http://orcid.org/0000-0003-4437-4517
Anil Ananthaneni Kumar http://orcid.org/0000-0003-0565-8752

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
