## [Reviewer comments · BMJ Open]

ARTICLE DETAILS

TITLE (PROVISIONAL)	Feasibility and acceptability pilot of Video-based Direct Observed Treatment (vDOT) for supporting antitubercular treatment in South India – A cohort study
AUTHORS	Rodrigues, Rashmi; Varghese, Suman; Mahrous, Mohammed; Ananthaneni, Anil; Ahmed, Mohammed; D'Souza, George

VERSION 1 – REVIEW

REVIEWER	Shibu, Vijayan PATH, TUBERCULOSIS
REVIEW RETURNED	07-Sep-2022

GENERAL COMMENTS	Treatment monitoring to treatment support Small sample size skews the proportions and may affect the statistical calculations – so interpreting cohort adherence score is tricky , however individual adherence score and acceptability is possible Digital literacy – roughly 52% had willingness in smartphone issues, need to options Among participants females were more is there a reason for this Standards of TB care in India 2014 – moved towards a adherence support system from DOTS, this may be captured in the background 99DOTS was rolled out before COVID , there is a statement in the manuscripts this was rolled out during COVID , which is incorrect It was not clear did the project paid for the internet There will be situations where people might have consumed but not recorded (due to many reasons) how this is accounted for Technical issues was notified as the reason for missing uploading, good to understand what issues were classified as technical The mean adherence among < 80% video acceptance is 51% , it will be difficult to have any Digital adherence tool (DAT) in isolation, this should not be positioned as a standalone tool, DAT is a quality improvement tool which helps in augmenting/ optimizing health workers involvement and helps in prioritizing who needs the right support, this article needs to frame the narrative in such away DAT may improve quality of service delivery Explain who else from the family /friend /community were counselled for supporting the patient Passcode for accessing the application looks complicated from the narrative , think about facial recognition or finger print , which is easy
---

REVIEWER	Fielding, Katherine London School of Hygiene & Tropical Medicine
REVIEW RETURNED	02-Dec-2022

GENERAL COMMENTS	Feasibility and acceptability of a video – based treatment adherence monitoring system in South India – A cohort pilot study (BMJopen-2022-065878) This is a small pilot study assessing the feasibility and acceptability of a video-supported therapy among 25 adult drug-sensitive TB patients enrolled from a referral teaching hospital in Bangalore, India. The video-supported therapy intervention was implemented by research assistants. Data are reported on treatment adherence, reasons for not uploading a video, patterns over calendar time for videos accepted/missed/rejected; and acceptability of the application and privacy data from the patient perspective. Abstract:  1. For the age description of age - I think ± 14 yrs refers to SD ? please clarify. 2. Avoid the terminology “statistically significant” – see guidelines for reporting P-values 3. Videos improved in quality and acceptability over time - suggest you make it clear this is calendar time; same comment for sentence in “conclusion” – also see my comment below on fig 2. Introduction:  4. Update WHO reference (#1) to the 2022 Global TB report. Methods:  5. Can you clarify the calendar time period of data collection? 6. Can you provide a reference for defining optimal adherence of $\geq 80\%$? 7. Please clarify if individual enrolled were aged ≥ 18 yrs or > 18 yrs? 8. Were all adult DS-TB patients diagnosed in the referral teaching hospital screened in a particular time period for this study? Or only selected patients screened? Results  9. Patient flow:  a) What were the reasons for being ineligible for the 258-150=108? b) In table 1 I suggest you separate out “not willing to learn to use smartphone” and “not interested in research”? c) From the text (page 7, line 32) can you separate out “not owning” and “not interested in learning to use a smart phone” d) 15/25 (60%) dropped out of the study within 2 months of enrolment – and were replaced. Was only one “main” reason documented for dropping out or could > 1 reason be recorded? 10. Table 2  a) 64% female – is this in keeping with sex ratio usually seen? b) Please add footnotes for acronyms BPL and APL c) 3/25 also dropped out at month 2 and 3 - what were the reasons? d) What is the purpose of the P-values in this table? For me, they are not necessary
---

e) Can a summary of the time from TB treatment start to when the participant was enrolled into the pilot study be added to this table

11. In Figure 2

a) please explain in the figure what n=2675 refers to

b) in the abstract it is described that “Participant videos improved in quality and a higher proportion met acceptability criteria over time” , based I assume on this graph. A similar description is not in the results section of the main text.

c) The x-axis of this graph is calendar time - which I guess may be of interest - but I also think a time scale of “time since enrolment” for each participant is of interest – with the hypothesis that as the participant gets used to using making/uploading the video you may see an improvement in the quality of the video?

12. Table 3: These are occasions when patients did not upload a video, not PwTB. Can the number of PwTBs also be added for each reason for not uploading? So that we can see, for example, there were 63 occasions where “claimed to have uploaded video” was the reason, from XX PwTB, etc.

13. It might be interesting to explore socio-demographic factors associated with not being contacted? Analysis could be done at the event level, but take into account patient-level clustering

14. Page 9 lines 11-14 – clarify the $\pm x\%$ refer to 95% CI? \pm SE? If \pm SE, why not rather show the 95% CI?

15. Table 4 if based on logistic regression of n=25 with outcome of “good adherence” (16/25)

a) For the cross-tabulations of each variable with the outcome I recommend you show the # with “good adherence”, the denominator (and %), and remove sub-optimal. For example for sex: for females 6/9 (67%) and males 10/16 (62 %). This the standard way of presenting such data.

b) It also needs to be made clear for the odds ratio, what is the baseline group. For sex the OR=1.20 compares odds of “good adherence” in females vs males. This is best made more clear by putting a 1 for the OR for “male”

c) My biggest concern with this analysis is the small sample size (n=25) – the adjusted analysis is not justified as likely to be over-parameterised. I would recommend this is removed. Using exact logistic regression does not help with this.

d) I strongly suggest the statement “that as none of the socio-demographic variable showed a significant significance” and then this “shows that demographic factors did not play a “significant role” in optimal adherence” should be rephrased. It is unlikely you have power to detect such factors with only 25 patients in the analysis. Absence of evidence is not the same as evidence of absence (see <https://www.bmj.com/content/311/7003/485> among

	many other papers in this issue). 16. Why were stacked bar charts used for figure 4 and not for figure 3? Seems sensible to keep this graphical display consistent for figures 3 & 4. 17. Clarify what is meant by “front and back end” - page 10, line 21. The data (69, 19%) relates to patient report so wouldn't their issues relate to front end issues (would they be aware of backend issues?) Discussion 18. Same comment as above (7d) regarding “other socio – demographic factors associated with optimal adherence” – see page 10, line 59/60 Comment on how widespread ownership of smart phone is.. 19. Page 11, lines 11-18: it seems that percentages are being compared across the current study with study reference 18. These percentages, however are measuring different things. The current study percentages refer to a summary of events (video not uploaded) and ref 18 percentages are for patients. 20. Comment that research assistants (not routine health staff) implemented the study – including uploading videos, reviewing uploaded videos and defined as accepted, rejected or missed. In routine practice this will be done by health staff and not a parallel cadre of staff.
--	---

VERSION 1 – AUTHOR RESPONSE

Reviewer 1		
1	Treatment monitoring to treatment support	Thank you for the comment. We have changed the title from ‘treatment monitoring’ to ‘treatment support’. It currently reads: “Feasibility and acceptability pilot of Video–based Direct Observed Treatment (vDOT) for supporting antitubercular treatment in South India – A cohort study”
2	Small sample size skews the proportions and may affect the statistical calculations – so interpreting cohort adherence score is tricky, however individual adherence score and acceptability is possible	Thank you for the comment. We agree that the small sample size will skew statistical calculations. However, the aim of the study was to assess the feasibility and acceptability of the vDOT intervention among PwTB. Through an exploratory approach. This study was only a pilot and not intended to identify statistically significant associations. Therefore, we did not calculate any sample size. To identify

		the effectiveness of the intervention an RCT is underway. We have addressed this as a limitation in the discussion. Page12; Line 29-31 “Overall, while the small sample size limits generalisability, the study has helped identify and address technical challenges with the application and while understanding its acceptability.”
3	Digital literacy – roughly 52% had willingness in smartphone issues, need to options	Thank you for the comment. In view of the low digital literacy rate, the participants were trained to use the basic features of the phone and the intervention application once they were enrolled in the study. We, therefore, were able to enroll participants, despite them not owning a smart phone. When participants found it challenging we ensured that a family member was counselled and trained to help the participant take the video and upload it on the application. Page 5; Line 24 – 25 “In case a participant as unable to use the application, a family member was chosen and trained to record and upload videos of the participant.”
4	Among participants females were more is there a reason for this	Thank you for the comment. Participants were enrolled consecutively (consecutive sampling was used). Hence, if it was a female patient who visited the enrollment center and consented to participate, we enrolled her. We did not intentionally enroll female participants over male participants. All the female participants in this study were literate. We do not think there was a specific reason for this trend. Page 6; Line 22 – 23 “Eligible male participants (62, 56%) who refused consent cited ‘time constraints’ while eligible female participants (48,44%) cited privacy and stigma related concerns as reasons for refusal.” Page 12 line 31 – 35 “Also, approximately 2/3rd of the participants were women, as many men

		with TB refused to participate in the study citing employment and data security issues. This implies a selection bias as the proportion of men with TB is higher than that of women(28) and so also their access to digital technologies. However, the valuable insight into the uptake of DAT by women, that the study provides, if given an opportunity, cannot be ignored.”
5	Standards of TB care in India 2014 – moved towards a adherence support system from DOTS, this may be captured in the background	Thank you for the comment. We have added a sentence on Page 3; Line 24 - 26. “Further, the Indian National TB Elimination Programme (NTEP), while adopting a patient centered approach in 2014 (8), also included mHealth solutions for adherence support.” This is followed by a description on the various mHealth interventions available.
6	99DOTS was rolled out before COVID, there is a statement in the manuscripts this was rolled out during COVID, which is incorrect	Thank you for the comment. 99DOTS was rolled out for patients with HIV-TB prior to the pandemic, however, 99DOTS was rolled out for patients with DSTB during the pandemic. We have rephrased the statements as follows: Page 3; Line 32 -34 ““99DOTS,” is another mHealth adherence support strategy that the NTEP, rolled out for patients with HIV-TB coinfection in 2015. The strategy subsequently covered all PwTB in 2021 during the Covid 19 pandemic, when DOTS posed a logistic challenge (13).”
7	It was not clear did the project paid for the internet	Thank you for the comment. We have included the following statement for clarity in the section in methods. Page 5; Line 28-29 “Each participant received INR 500/ month for Internet time as well as study related travel.”

8	There will be situations where people might have consumed but not recorded (due to many reasons) how this is accounted for?	Thank you for your comment. We agree that there may have been instances where participants have taken the dose but have not uploaded the video. If vDOT as an intervention has to be feasible as per this study, the intervention itself may not be considered if the doses cannot be viewed. For the purpose of this study, any dose that has not been video recorded and uploaded is considered 'missed' unless proven otherwise. We have added a statement in the discussion that reads as follows: Page 11; Line 20-24 "However, for vDOT, if we did not receive a video we equated it to a missed dose. While doing so underestimated the true adherence it likely ensured robustness. This especially, as the viewing every dose taken via the mobile application, is the alternative to DOT that we propose. While this might imply vDOT as a standalone intervention is questionable, evidence indicates that DAT such as vDOT holds promise when combined with other interventions such as family/community DOTS or peer support."
9	Technical issues was notified as the reason for missing uploading, good to understand what issues were classified as technical	Thank you for the comment. We have detailed the technical issues faced by, both, the participants and the operators have been listed in table 6. Some of the issues that the participants faced had to do with pop-up reminders and passcodes. At the back end we faced issues with reshuffling, lag between uploading and viewing the videos and incorrect calculations of adherence rates to name a few.
10	The mean adherence among < 80% video acceptance is 51%, it will be difficult to have any Digital adherence tool (DAT) in isolation, this should not be positioned as a standalone tool, DAT is a quality improvement tool which helps in	Many thanks for the comment. We have included the suggestion under the sub-heading 'discussion'. Page 11; Line 20-24

	augmenting/optimizing health workers involvement and helps in prioritizing who needs the right support, this article needs to frame the narrative in such way DAT may improve quality of service delivery	“However, for vDOT, if we did not receive a video we equated it to a missed dose. While doing so underestimated the true adherence it likely ensured robustness. This especially, as the viewing every dose taken via the mobile application, is the alternative to DOT that we propose. While this might imply vDOT as a standalone intervention is questionable, evidence indicates that DAT such as vDOT holds promise when combined with other interventions such as family/community DOTS or peer support.”
11	Explain who else from the family /friend /community were counselled for supporting the patient	Thank you for the comment. We did not train any specific person from the family or community. This intervention was superimposed on the NTEP activities and the DOTS center staff took care of counselling and advising family/community members as the case may be. However, our research assistants called the concerned participant for each missed dose and the participant visited the center once a month in person for follow-up as well. This is over and above the daily medication reminders programmed into the application.
12	Passcode for accessing the application looks complicated from the narrative, think about facial recognition or finger print, which is easy	Thank you for the comment. Our participants had good numerical literacy even if their language literacy was not good. However, we do agree that the passcode security feature can prove to be a bit of a hassle. The facial recognition and fingerprint options are expensive. The current version of the intervention was designed to be operated on a basic android model. In further revisions of the application these options, if they become inexpensive, can be made available.
Reviewer 2		
1	For the age description of age - I think ± 14 yrs refers to SD? please clarify	Thank you for the comment. Yes, it refers to the standard deviation, we have re-worded it as ‘The mean \pm SD age of the participants was 33 \pm 14 years’. Page 2; Line 15

2	Avoid the terminology “statistically significant” – see guidelines for reporting P-values	Thank you for the comment. We have carried out a panel analysis and the statement in the results section of the abstract as follows (Page 2; Line 21- 22): “Mixed model logistic regression showed that those who are married are more likely have daily adherence to ATT.”
3	Videos improved in quality and acceptability over time - suggest you make it clear this is calendar time; same comment for sentence in “conclusion” – also see my comment below on fig 2.	Thank you for the comment. As suggested in the comments below, we have looked at the patterns of video quality as time after enrollment. Thus, we are not referring to calendar time any longer.
4	Update WHO reference (#1) to the 2022 Global TB report.	Thank you for the comment. In the introduction we have made the suggested change as follows (Page 3; Line 2-3): “Globally, 6.4 million people were diagnosed with tuberculosis in 2021 with India accounting for approximately a third of these cases.”
5	Can you clarify the calendar time period of data collection?	Thank you for the comment, we have added a statement in the Methods section (Page 4; Line 3 - 4). “The participants for this study were enrolled consecutively from those PwTB who were referred to the DOTS center for treatment between July 2017 and November 2018.”
6	Can you provide a reference for defining optimal adherence of $\geq 80\%$?	Thank you for the comment. We have added a reference (reference 16). Nahid P, Dorman SE, Alipanah N, Barry PM, Brozek JL, Cattamanchi A, et al. Official American Thoracic Society/Centers for Disease Control and Prevention/Infectious Diseases Society of America Clinical Practice Guidelines: Treatment of Drug-Susceptible Tuberculosis. Infect Dis Soc Am. 2016;63.

7	Please clarify if individual enrolled were aged ≥ 18 yrs or >18 yrs?	Thank you for the comment. We have amended the statement in the methods section as follows: (Page 4; Line 5 -7) “The participants were recruited comprised newly diagnosed patients with drug sensitive TB, aged ≥ 18 years, with at least 2 months of treatment pending at enrolment.”
8	Were all adult DS-TB patients diagnosed in the referral teaching hospital screened in a particular time period for this study? Or only selected patients screened?	Thank you for the comment. The hospital has a government run DOTS center attached to it. All patients who were referred to the DOTS center for treatment were screened for this study. However, there are other DS-TB patients who would have received care by physicians at the hospital and these patients were not screened for our study. For clarity we have added the following statement: (Page 4; Line 3 - 4). “The participants for this study were enrolled consecutively from those PwTB who were referred to the DOTS center for treatment between July 2017 and November 2018.”
9	Patient flow: a) What were the reasons for being ineligible for the 258-150=108? b) In table 1 I suggest you separate out “not willing to learn to use smartphone” and “not interested in research”? c) From the text (page 7, line 32) can you separate out “not owning” and “not interested in learning to use a smart phone” d) 15/25 (60%) dropped out of the study within 2 months of enrolment – and were replaced.	Thank you for the comment. We have made the following amendments: a) The reasons for ineligibility are as follows: >60 years of age:40; <18years:17; Category 2 ATT: 19; PwHIV-TB: 23; On injectable antibiotics: 3; MDR-TB: 3; XDR-TB:1; <2 months of treatment remaining: 1; Extremely sick:1. In the Results section we have included the following statement: (Page 6; 24 -26) “The most common reasons for ineligibility were age <18 years or >60years (40, 37%), patients with HIV-TB coinfection (23, 21%) and recurrent TB (2 anti-tubercular therapy) (19, 17%).” b) We have separated out “not willing to learn to use smartphone” and “not interested in research”. c) These are not necessarily mutually exclusive reasons, participants have

	Was only one “main” reason documented for dropping out or could >1 reason be recorded?	cited these two reasons simultaneously. d) Yes, participants have on occasion cited more than one reason for dropping out of the study, we have selected one main reason to be listed in Table 1.
10	Table 2 a) 64% female – is this in keeping with sex ratio usually seen?	Table 2 a) Thank you for the comment. Participants were enrolled consecutively (consecutive sampling was used). Hence, if it was a female patient who visited the enrollment center and consented to participate, we enrolled her. We did not intentionally enroll female participants over male participants. All the female participants in this study were literate. We do not think there was a specific reason for this trend. Page 6; Line 22– 23 “Eligible male participants (62, 56%) who refused consent cited ‘time constraints’ while eligible female participants (48,44%) cited privacy and stigma related concerns reasons for refusal.” Page 12 line 31 – 35 “Also, approximately 2/3rd of the participants were women, as many men with TB refused to participate in the study citing employment and data security issues. This implies a selection bias as the proportion of men with TB is higher than that of women(28) and so also their access to digital technologies. However, the valuable insight into the uptake of DAT by women, that the study provides, if given an opportunity, cannot be ignored.” b) Thank you for the comment. A footnote has been added for table 2 with the expansions for the acronyms APL and BPL. c) Thank you for your comment. All three participants withdrew consent. We have modified the footnote in table 2, it currently reads, ‘Withdrew consent:3, 2nd month 2 (due to family concerns regarding stigma), 3rd month 1 (due to other health related issues)’ d) Thank you for the comment. The p-value column for table 2 has been deleted.

	b) Please add footnotes for acronyms BPL and APL c) 3/25 also dropped out at month 2 and 3 - what were the reasons? d) What is the purpose of the P-values in this table? For me, they are not necessary e) Can a summary of the time from TB treatment start to when the participant was enrolled into the pilot study be added to this table	e) Thank you for the comment. A statement has been made in the results section which reads as follows: (Page 6; Line 23 - 24) “The median time to enrollment from treatment initiation was 1day (IQR 0-15 days) with a maximum of 109 days.” We have also added this variable in Table 2 which describes the demography and clinical features separated by sex.
11	In Figure 2 a) please explain in the figure what n=2675 refers to	In Figure 2 (Now figure 3), we have limited the data to after 6 months of exposure, as only 3 participants had more than 6 months of exposure and it misrepresents the findings of the study. a) Thank you for the comment. We have made changes in the text box as follows: ‘Total number of videos expected =3193 Total number of videos received =2501’ The numbers have changed as we have presently included only the 25 participants who completed the study and included only the first six months of treatment. b) Thank you for the comment. We have added a statement in the text under the results section. Page 8; Line 4-6 “The proportion of uploaded videos that met the acceptability criteria

	b) in the abstract it is described that “Participant videos improved in quality and a higher proportion met acceptability criteria over time”, based I assume on this graph. A similar description is not in the results section of the main text. c) The x-axes of this graph is calendar time - which I guess may be of interest - but I also think a time scale of “time since enrolment” for each participant is of interest – with the hypothesis that as the participant gets used to using making/uploading the video you may see an improvement in the quality of the video?	increased over time as participants’ familiarity with the application grew. (Fig. 3)” c) Thank you for the comment, as per your suggestions we have changed the x-axis from calendar time to time since enrollment.
12	Table 3: These are occasions when patients did not upload a video, not PwTB. Can the number of PwTBs also be added for each reason for not uploading? So that we can see, for example, there were 63 occasions where “claimed to have uploaded video” was the reason, from XX PwTB, etc.	Thank you for the comment. As per your suggestion we have added a new column in ‘Table 3 (Now table 5)’ with the number of participants who cited that particular reason for not uploading a video and the corresponding percentage in parenthesis.
13	It might be interesting to explore socio-demographic factors associated with not being contacted? Analysis could be done at the event level, but take into account patient-level clustering	Thank you for the comment. As suggested, we have carried out a mixed model analysis and the tables are attached in annexures. Page 8; Line 15 - 17 “A mixed model analysis was carried out with responsiveness as the outcome variable. The ICC for the null model indicates that 70% of the variation is accounted for by inter cluster variation. The model indicated that those who were unemployed were more likely to have daily responsiveness during the follow – up period.”
14	Page 9 lines 11-14 – clarify the $\pm x\%$ refer to 95% CI? $\pm SE$? If $\pm SE$, why not rather show the 95% CI?	Thank you for the comment. The $\pm x\%$ referred to SD. However, as per your suggestion we have amended the paragraph as follows. Page 8; Line 7 - 10 “The overall average adherence was 70% (95%CI: 68%-87%), and ranged between 12-99%. Sixteen (64%) participants were optimally adherent through vDOT of $\geq 80\%$ [mean adherence (95% CI) 91% (88%-

		95%]). The remainder had less than 80% of videos accepted [12 participants, mean (95% CI) adherence 51% (38% - 68%).”
15	Table 4 if based on logistic regression of n=25 with outcome of “good adherence” (16/25) a) For the cross-tabulations of each variable with the outcome I recommend you show the # with “good adherence”, the denominator (and %), and remove sub-optimal. For example for sex: for females 6/9 (67%) and males 10/16 (62 %). This the standard way of presenting such data. b) It also needs to be made clear for the odds ratio, what is the baseline group. For sex the OR=1.20 compares odds of “good adherence” in females vs males. This is best made more clear by putting a 1 for the OR for “male” c) My biggest concern with this analysis is the small sample size (n=25) – the adjusted analysis is not justified as likely to be over-parameterised. I would recommend this is removed. Using exact logistic regression does not help with this. d) I strongly suggest the statement “that as none of the socio-demographic variable showed a significant significance” and then this “shows that demographic factors did not play a “significant role” in optimal adherence” should be rephrased. It is unlikely you have power to detect such factors with only 25 patients in the analysis. Absence of evidence is not the same as evidence of absence (see https://www.bmj.com/content/311/7003/485 among many other papers in this issue).	Thank you for the comment (Now table 3). a) We have modified the column with optimal adherence in table 4 as per your suggestion. b) We have specified the reference group in the unadjusted OR. c) We agree with your comment, however, we wish to represent this information to the readers. We have carried out a mixed model analysis (Table 5). d) We have modified the statement in the results section as suggested: Page 8; Line 11 - 13 “Participants who were female, those married, those literate, those gainfully employed, those below poverty line socio-economic status, those who were enrolled earlier on in their treatment and those who preferred DOT were more adherent to medication (Table 3). However, the analysis lacks power.”
16	Why were stacked bar charts used for figure 4 and not for figure 3? Seems sensible to keep this graphical display consistent for figures 3 & 4.	Thank you for the comment we have made the chart type uniform in fig 3 (now fig 4) & 4 (now fig 5).
17	Clarify what is meant by “front and back end” - page 10, line 21. The data (69, 19%) relates to patient report so wouldn't their issues relate to front end issues (would they be aware of backend issues?)	Thank you for the comment, we have made the following changes. In the methods section, under the sub-heading ‘The vDOT mobile Application’, we have specified the meaning replaced the term ‘web interface’ with ‘provider interface’.

		In the results section we have made the following amendment for clarity: Page 10; Line 21-23 “The most common reason for a video not uploaded or recorded as uploaded but not received by the study team was “technical issues with the application” this referred to issues at both, the user and provider interface (89, 19%).”
18	Same comment as above (7d) regarding “other socio – demographic factors associated with optimal adherence” – see page 10, line 59/60 Comment on how widespread ownership of smart phone is.	Thank you for the comment. We have amended the statement as follows: Page 11; Line 2 “We observed that nearly two – thirds (16, 64%) of the participants had optimal adherence (i.e., ≥80%) with vDOT, with no sociodemographic factors influencing adherence.” Thank you for the comment we have included the following statement in the discussion section: Page 3; Line 20-21 “Especially as India’s smartphone penetration is 93% (91% rural and 97%) (6).”
19	Page 11, lines 11-18: it seems that percentages are being compared across the current study with study reference 18. These percentages, however are measuring different things. The current study percentages refer to a summary of events (video not uploaded) and ref 18 percentages are for patients.	Thank you for the comment. We agree with your observation and have made the following amendment to the statement in the results section: Page 11; Line 13-17 “Of the 2501 doses received on the application, over 90% were of acceptable quality, While 40% of the participants forgot to record a video at least once during follow-up, only 12% had actually missed taking their medication. A cross-sectional study done among HIV-TB co-infected patients using 99DOTS, reported that 26% had forgotten to take the medication and 6% participants forgot to make a call after taking the medication (20).”
20	Comment that research assistants (not routine health staff) implemented the study – including uploading videos, reviewing uploaded videos and defined as accepted, rejected or missed. In	Thank you for the comment.

	routine practice this will be done by health staff and not a parallel cadre of staff.	As suggested we have included the following statement in the description of the intervention: Page 4; Line 39 - 42 “Trained research assistants (not routine health staff) reviewed the uploaded videos at the provider interface of the application and classify the videos as ‘accepted’ (dose taken) or ‘rejected’ (video received but dose not taken). The videos were accepted if the participant was identified, all the pills were shown before swallowing under adequate lighting. If not, the video was ‘rejected’.”
--	--	--